# Impact of tides and sea-level on deep-sea Arctic methane emissions

Nabil Sultan [1✉], Andreia Plaza-Faverola [2], Sunil Vadakkepuliyambatta [2], Stefan Buenz[2] & Jochen Knies[2,3]

Sub-sea Arctic methane and gas hydrate reservoirs are expected to be severely impacted by ocean temperature increase and sea-level rise. Our understanding of the gas emission phenomenon in the Arctic is however partial, especially in deep environments where the access is difficult and hydro-acoustic surveys are sporadic. Here, we report on the first continuous pore-pressure and temperature measurements over 4 days in shallow sediments along the west-Svalbard margin. Our data from sites where gas emissions have not been previously identified in hydro-acoustic profiles show that tides significantly affect the intensity and periodicity of gas emissions. These observations imply that the quantification of present-day gas emissions in the Arctic may be underestimated. High tides, however, seem to influence gas emissions by reducing their height and volume. Hence, the question remains as to whether sea-level rise may partially counterbalance the potential threat of submarine gas emissions caused by a warmer Arctic Ocean.

[1] Ifremer, Département REM, Unité des Géosciences Marines, F-29280 Plouzané, France. [2] Centre for Arctic Gas Hydrate, Environment and Climate, Department of Geosciences, UiT-The Arctic University of Norway in Tromsø, Tromsø, Norway. [3] Geological Survey of Norway, NO-7491 Trondheim, Norway. ✉email: nabil.sultan@ifremer.fr

Although ocean methane emissions are considered to be widespread[1–6] their dynamics and the physical processes behind their evolution are little understood. Given the impact of methane as a greenhouse gas, the dynamic of oceanic methane emissions, which could potentially reach the atmosphere, introduces a non-negligible doubt on the global budget of atmospheric methane[7].

Gas emissions are usually caused by natural active geological processes and are the expression of seabed and sub-seabed features (e.g., faults, fractures, pockmarks, mud volcanoes, hydrothermal vent complexes). Gas emissions in the water column are often the result of gas exsolution during fluid ascent[8], migrating sub-seabed gas accumulation through fractures[9,10] and dissociating gas hydrates through changes in thermodynamic conditions[11–13]. Trigger and periodicity of gas emissions are mainly controlled by stress accumulation (source), strains, and fractures (paths). For instance, gas emissions and methane leakage appear to be correlated with the tectonic stress field and the distribution of micro-seismicity along the North Anatolian Fault in the Sea of Marmara[3,14] and along the Vestnesa Ridge[15].

Environmental changes in the Arctic are recently highlighted as an additional factor affecting ongoing gas emissions. From multi-year hydro-acoustic surveys off western Svalbard, (2008–2014) it is evident that seasonal changes in ocean temperature may control the "spatial migration" of methane seepage[4,16]. Moreover, long-term (over 1 year) monitoring data from the NEPTUNE cabled observatory offshore Vancouver Island confirm that gas emissions are controlled by tide cycles[17]; an explanation that has also been suggested for shallow-sea active gas emissions on the west Svalbard continental shelf[4].

Hence, several decades of research on ocean methane emissions have demonstrated the sensitivity of such a dynamic system to geological[18] and environmental[19] variables on short and long-term scales. However, there is no consensus on how gas emissions from these dynamic systems fit into a climate-change scenario[20]. For example, while field observations on formerly glaciated margins point towards massive emissions of greenhouse gases following ice-sheet retreat[21], recent analysis of ice-cores suggest that post-LGM (Last Glacial Maximum, ~20,000 years ago) methane release from old reservoirs was too small to impact the global climate[22]. A challenging aspect on the quantification of present-day emissions is that seepage may be more widespread and abundant than we are able to identify based on the presence of gas plumes in hydro-acoustic surveys. The current state of knowledge does not allow us to define a concrete outline on the impact of future temperature and sea-level rises on methane bubble emissions[23].

In this study, we analyze an element of this phenomenon by hypothesizing that even small changes in sea-level may have a global impact on the intensity of deep-sea gas emissions. We suggest that tides can be used as a proxy to predict and quantify variability in the amount of gas released on a daily basis. To test our hypothesis, we carried out in-situ sediment pore-pressure and temperature measurements over 4 days on a widely investigated methane seepage system in the Arctic, the Vestnesa Ridge (NW Svalbard). We aim at better characterizing the short-term periodicity of deep-marine seepage and the effect of tides on the pressure field that controls the emissions.

## Results and discussion

### Seafloor seepage off western Svalbard and study sites.
Gas plumes in the water column have been identified in hydroacoustic data at discrete locations all along the western Svalbard margin[24]: (1) at ~80 m water depth unrelated to present day gas hydrates[25,26], (2) at ~300–400 m water depth near the shelf break and the gas hydrates stability zone pinch-out[18,27–29] and (3) at ~1200 m water depth on the eastern segment of the Vestnesa Ridge[5,30–33], a contourite drift of ~100 km length and 30 km width (Fig. 1)[32].

The seepage area with the highest density of plumes is situated at 300–400 m water depth near the shelf break. Here seepage has been associated with the dissociation of gas hydrates either by seasonal and larger term (thousands of years) temperature changes[18,27,29] or by a decrease in overburden pressure following the ice-sheet retreat since the LGM[28,34]. On Vestnesa Ridge in ~1000–1200 m water depth, active gas plumes appear restricted to its easternmost part[31]. The presence of pockmarks, seemingly inactive, on the western Vestnesa Ridge segment indicates past seepage (Fig. 1). However, the lack of observation of gas plumes in the water column does not necessarily indicate that the system is entirely sealed and inactive at present[35]. Micro-seepage may occur with an intensity (or periodicity) that has not yet been visible on hydro-acoustic profiles.

In this study we report on the pore pressure and temperature data measured with a piezometer deployed from RV "Kronprins Haakon" at two stations (Fig. 1 and Table 1)[36]. Station PZM1 is located on the continental slope at the southeastern onset of Vestnesa Ridge, next to an elongated seafloor depression and associated sub-seabed fluid migration feature (Supplementary Figs. 1, 2). The site is well within the gas hydrate stability zone and free gas and gas hydrates have been documented from gravity cores from the depression[37] (Supplementary Figs. 3 and 4). Station PZM2 is located ~80 km westward from station PZM1, on the western segment of the Vestnesa Ridge (Supplementary Fig. 5). Both piezometers were deployed at sites where gas plumes had not been observed in hydro-acoustic data available from the area (Supplementary Figs. 6 and 7). We explicitly selected the sites to avoid active geological structures while characterizing the hydraulic and thermal properties of the near-surface sediment.

### Pore pressure and temperature data.
The piezometer is designed to carry-out in situ pore-pressure and temperature measurements (see methods paragraph). Piezometer penetration compresses and shears the surrounding sediments under undrained conditions, thus generating excess pore-water pressure and heat development. Once piezometer insertion stops, penetration-induced pressures and temperatures dissipate monotonically with time allowing to characterize the in-situ hydraulic and thermal regimes.

Pore pressure and temperature data from PZM1 recorded over 3 days are shown in Fig. 2. The temperature decays to reach in-situ equilibrium temperature in approximately four hours (Fig. 2). The temperatures at the deepest eight sensors are almost constant throughout the monitoring period whilst the uppermost sensor recorded an important deviation towards warmer temperatures. Pore pressure data from the nine pore-pressure sensors fluctuate throughout the monitoring period (Fig. 2). Negative pore pressures as low as −240 kPa were recorded by the shallowest sensor.

PZM2 measured pore pressure and temperature over more than 4 days (Fig. 3). Temperature data from the two upper sensors show subtle fluctuations during the recording period. After the decay of the piezometer penetration-induced pressures, pore pressure data recorded by the upper four sensors show some pressure perturbations initiated mostly during low tide periods (Fig. 3b).

### Seawater–Sediment Interactions.
Piezometer pore pressure and temperature measurements allow studying the dynamics of free gas at unprecedented resolution and time scales across the

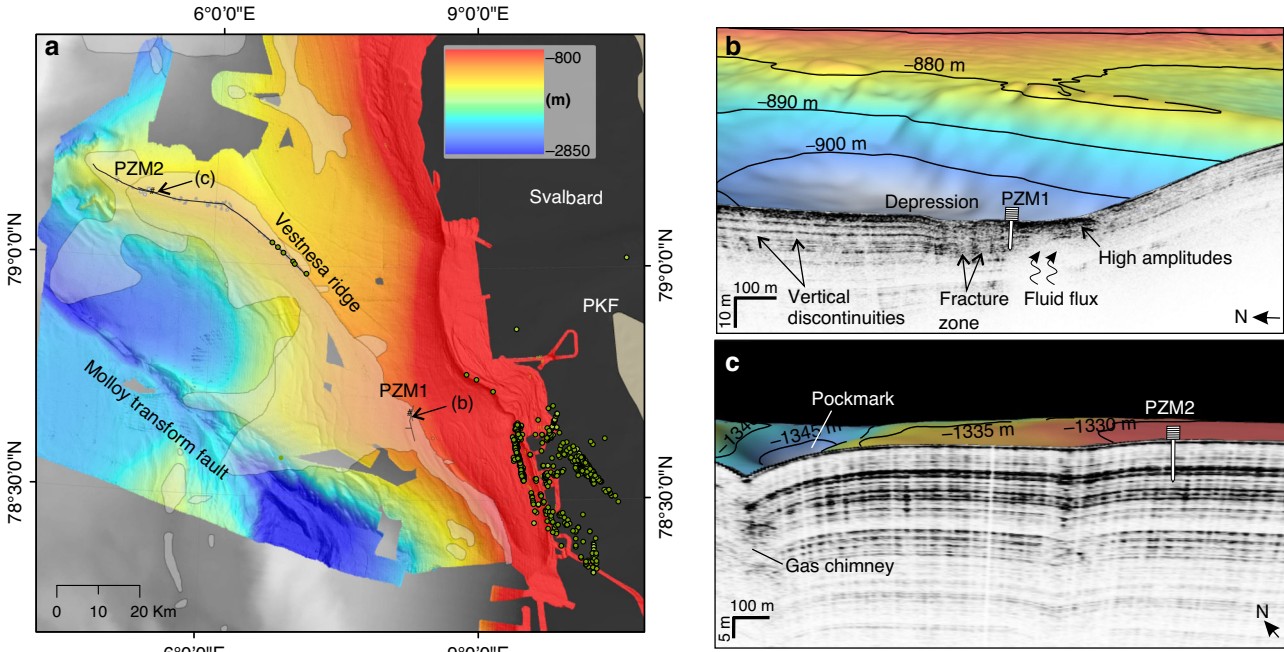

**Fig. 1 Study area and distribution of seafloor seepage. a** Location of investigated sites off the western Svalbard margin. Background bathymetry from IBCAO[55] (gray); higher resolution bathymetry from UiT—The Arctic University of Norway database. The distribution of surveyed gas plumes during yearly expeditions to the area is indicated by green circles. Pockmarks without associated gas plumes on the western Vestnesa Ridge are indicated as gray dots. Shaded areas over the bathymetry correspond to mapped bottom simulating reflectors[56]. **b** Station PZM1 is located within the gas hydrate stability area and near a sediment depression where gas hydrates have been recovered[37]. **c** PZM2 was deployed in an area characterized by a seismic facies showing parallel reflections and no major vertical discontinuity.

**Table 1 Piezometer stations.**

| Simplified names | Sites | # of sensors | Coordinates | Water depth [m] | Length [m] | Recording period |
|---|---|---|---|---|---|---|
| PZM1 | KH-01-PZM1 | 9 | 78.687 N–8.256 E | ~910 | 8.2 | 22/10/2019—3:54 25/10/2019—10:40 |
| PZM2 | KH-05-PZM2 | 9 | 79.143 N–5.274 E | ~1330 | 9.92 | 26/10/2019—12:11 31/10/2019—2:08 |

Characteristics of piezometers used in the present study.

western Svalbard margin. Sediment pore pressures are measured relative to hydrostatic pressure using differential pressure gauges connected to the open seawater (see methods paragraph and Supplementary Fig. 8). In this context, measured differential pore pressure during a change in sea level becomes an indicator of the type of fluids saturating the sediment as described below:

For seawater fully saturating sediment pores, a change in the height of the seawater column (i.e., tides) is expected to affect both sides of the differential sensor equally without any change in the recorded pore pressure.

In the presence of free gas within the sediment pores, a change in the height of the seawater column is likely to affect the measured differential pore pressure because of the high compressibility of the free gas with respect to the seawater[38,39]. A drop in hydrostatic pressure related to tides reduces the total stress and thus causes free gas expansion and exsolution[40], thereby increasing the compressibility of the sediment pore fluid and disturbing the measured differential pore pressure[41]. Free gas emissions can be also captured by differential pore pressure gauges since the sensors measure the difference between hydrostatic pressure related to seawater column height

(water depth) and sediment pore pressure. A negative differential pore pressure (pressure lower than the hydrostatic pressure) is, therefore, a strong indicator of the presence of low-density fluid saturating the sediment pores and migrating into the water column.

**Evidence for the occurrence of free gas**. Piezometers PZM1 and PZM2 were deployed on supposedly inactive sites in terms of gas emissions in the water column (Fig. 1 and Supplementary Figs. 6 and 7). However, PZM1 is located ca. 1.5 km north from a gravity core where gas hydrate nodules were sampled at 1.5 mbsf (Supplementary Figs. 2 and 3). Both gravity core with hydrates and piezometer station are located along an elongated depression that contains smaller depressions about 300–400 m wide (Supplementary Fig. 1). Acquired sub-bottom profiles indicate the presence of transparent facies that are typically associated with the occurrence of gas and gas hydrate (Fig. 1b). In comparison, sub-bottom profiles along the western Vestnesa Ridge segment show continues reflections and homogenous facies in the surrounding of PZM2 (Fig. 1c).

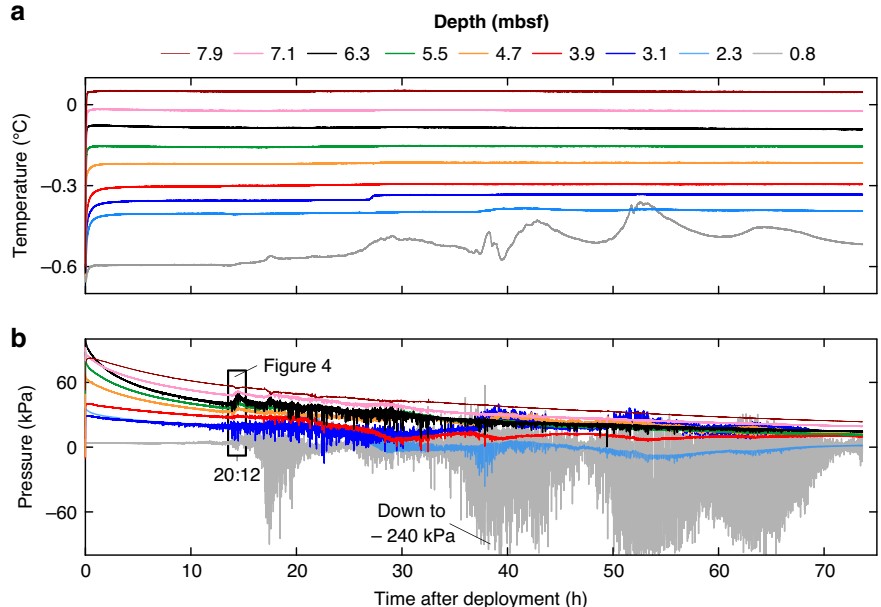

**Fig. 2 Data from piezometer site PZM1. a** Temperature and **b** pore pressure versus time. The different colors indicate the depth below the seabed. Sensor depths are between 0.8 mbsf (gray curve) and 7.9 mbsf (brown curve). Pressure axis limited to −100 kPa, for better visualization of the majority of the data.

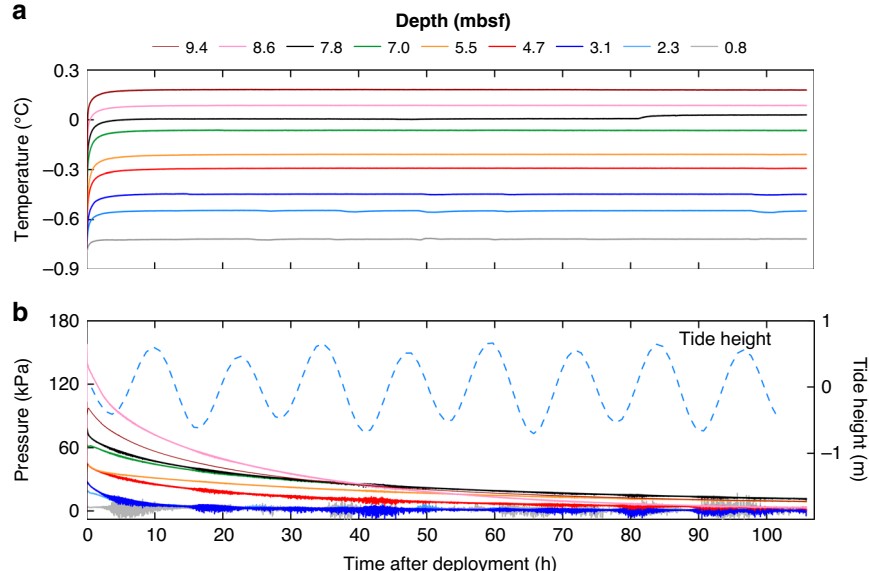

**Fig. 3 Data from piezometer site PZM2. a** Temperature and **b** pore pressure versus time. The different colors indicate the depth below the seabed. Sensor depths are between 0.8 mbsf (gray curve) and 9.4 mbsf (brown curve). Tidal heights obtained at the piezometer location from the TPXO 9.0 global tidal model[46,47] are shown as dashed light blue line.

Positive pore pressure data and the flat temperature trend from site PZM2 are consistent with the lack of gas plumes in water column acoustic data. However, the pressure profile suggests the presence of free gas partially saturating the shallow sediments (between 0.8 and 4.7 mbsf, Fig. 3). The four shallowest differential pressure sensors of PZM2 showed a response to tidal cycles indicating the presence of sediment gas-charged fluid. Since the amplitude of the pore pressure fluctuations is proportional to the gas content[41], data from PZM2 (Fig. 3) indicate a decrease of the gas content with depth. Under high tides, the absence of pore pressure fluctuations indicates the total dissolution of free gas. The absence of any temperature fluctuations accompanying the pressure fluctuations recorded by the upper four sensors of

PZM2 suggests the absence of significant upward fluid advection carrying hotter fluid. Pore pressure data recorded by PZM2 are typical of partially gas-saturated sediments. Piezometric data acquired from several comparable environments show similar characteristics[38,42,43]. Data presented in Fig. 3 demonstrate that shallow pore fluids at site PZM2 contain dissolved gas that exsolves under low tides. Nevertheless, the excess pressure generated by exsolved gases is not enough to overcome the sediment strength and in-situ minor principal effective stresses and thus prevents upward advection of free gas.

Data from PZM1 exhibit pore pressure fluctuations along with corresponding temperature fluctuations in the upper-most sensor. Temperature peaks above the measured seawater

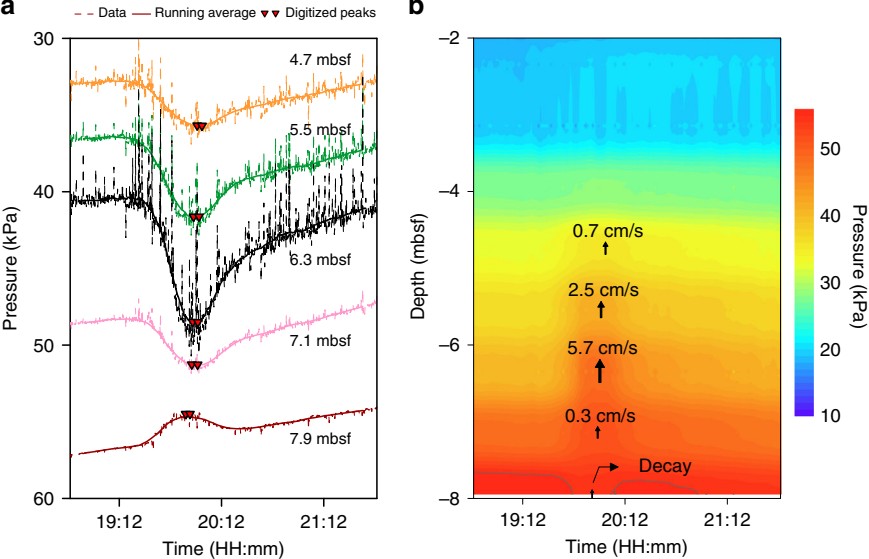

**Fig. 4 Gas bubble velocities. a** Pore-pressure measurements on the five lowermost sensors in the sediments measured at site PZM1 (record timing in Fig. 2). **b** Rising velocity of gas bubbles calculated from pore-pressure fluctuations in (**a**) projected on pore pressure contours at site PZM1.

| Time range of peaks after deployment (s) | Depth (mbsf) | Calculated mean velocity (cm/s) |
|---|---|---|
| 51757–51906 | 7.9 | – |
| 51972–52175 | 7.1 | 0.3 |
| 52004–52184 | 6.3 | 5.7 |
| 52032–52223 | 5.5 | 2.5 |
| 52161–52331 | 4.7 | 0.7 |

Table 2 Calculation of gas bubbles velocities.

Calculated mean velocities at site PZM1 from the pore pressure recorded between 19:12:00 and 21:12:00.

temperature (i.e., −0.64 °C) in the upper sensor indicate upward fluid flow (Fig. 2a). The measured pressure fluctuation reflects the effect of changes in hydrostatic pressure on compressible fluid (i.e., gas) present in the pore space. Pore pressure noise-like data are an indicator of the presence of free gas within the sediment pores while a negative pore pressure is an indicator of upward free gas migration into the water column.

**Rising velocity of gas bubbles**. Pore-pressure fluctuations at PZM1 are most evident on the five lowermost sensors, where a pressure increase is recorded on all sensors except the deepest one at 7.9 mbsf, which recorded a pressure decrease (Fig. 4a). Such a pressure profile can be explained by gas bubbles rising through the sediment with pore-pressure decay at the rear of the gas and pore pressure increases on the bubble gas front (Figs. 2 and 4a). Rising velocity of gas bubbles was calculated using the distance between consecutive sensors divided by the time interval between two successive maximum pore pressures (Fig. 4a and Table 2). Data shows a rising velocity of between 0.3 cm/s and 5.7 cm/s (Fig. 4b). Near the seabed, the pore pressure front becomes more diffusive, preventing the calculation of a bubble rising velocity.

The negative pore pressures measured by the first upper sensor of PZM1 (i.e., during the entire period shortly after deployment) may be explained by the presence of free gas accumulations saturating the sediment and expulsing free gas in the water column at this location. From the negative pressure cycles recorded by the uppermost sensor at site PZM1 (Fig. 2), it is

possible to infer five intermittent events of bubbles rising through the water column (Fig. 5).

The height ($h$) of a potential gas column over the sensor during these events can be estimated using the negative differential pore pressures ($\Delta P$) measured by the sensors and gas ($\rho_g$) and pore water ($\rho_w$) unit weights (Eq. (1)):

$$h = \frac{\Delta P}{\rho_w - \rho_g} \qquad (1)$$

The unit weight of methane ($\rho_g$) for the considered pressure and temperature can be as low as 0.66 kN/m$^3$. Assuming continuous gas columns above the sensor every time a gas emission pulse takes place, the equivalent gas plume can be up to 25 m high (Fig. 5). The equivalent continuous gas height must be considered as a lower limit since an alternation between gas bubbles and seawater above the seabed would require a much higher gas plume to explain the same measured negative pressure. Temperature data recorded at the same level (red curve in Fig. 5) also shows curve fluctuations with four to five events fitting with gas plume heights. The highest inferred gas plumes fit with the highest recorded temperatures. Technically, the gas flow accompanying the 25 m high zone of gas bubbles would produce a density contrast sufficiently strong to be seen as a gas plume in sonar data. However, our acoustic surveys (Supplementary Figs. 6 and 7) didn't show any presence of free gas in the water column suggesting, as it was shown in the case of the Cascadia margin[44] and the continental margin off SW Taiwan[45], that timing of survey is an important factor determining why seepage at PZM1 site has not been spotted in sonar profiles.

**Tides as a forcing mechanism**. Upward gas-charged fluid migration was suspected to be responsible for the recorded negative differential pressure cycles and temperature fluctuations measured by the upper sensor of PZM1. To check this observation, we conducted numerical calculations using a 1D transient diffusion-advection heat transfer model (see methods). The considered boundary conditions are an imposed temperature at the base of the model domain equivalent to the measured temperature at 2.3 mbsf and a constant seabed temperature equal to −0.64 °C. Although tidal currents may induce seabed

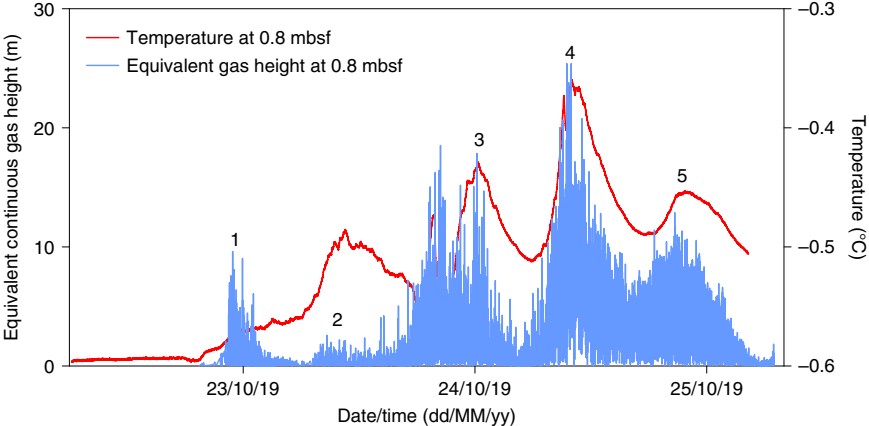

**Fig. 5 Gas plume heights.** Equivalent continuous gas plume heights versus time (blue curve) derived from pore pressure measured at 0.8 mbsf. The plume height peaks (indicated by numbers in the figure) coincides with temperature peaks measured at the same level (red curve).

temperature fluctuations, temperatures from the upper sensor of PZM2 remained almost constant during the 4-day monitoring period indicating that that the impact of currents on the studied process was negligible. The thermal diffusivity was derived from the temperature decay after piezometer installation and was taken as constant and equal to $3.5 \times 10^{-5}$ m²/s.

The upward fluid velocity $v_z$ was determined from the hydraulic gradient measured between the two shallowest sensors (0.8 and 2.3 mbsf) and by defining a hydraulic conductivity equal to $2.5 \times 10^{-5}$ m/s to fit with the measured rising velocity of gas bubbles at 4.7 mbsf shown in Fig. 3. The predicted temperature field (Fig. 6a) indicates significant temperature fluctuations on the second and third days of measurement. Comparison between the measured and predicted temperature at 0.8 mbsf (Fig. 6b) shows that three of the four temperature peaks were reproduced suggesting that vertical fluid advection may explain the measured temperature fluctuations. Moreover, the duration of each thermal pulse can be compared to the period of tide cycles. The three main upward fluid events have a duration of 12 h each (Fig. 6c). Tidal height and currents derived at the piezometer location from the TPXO 9.0 global tidal model[46,47] illustrate that the lowest eastward tide velocity ranges (lower than −3 cm/s) at low tide periods correspond in time to temperature and upward velocity peaks (Fig. 6d). These results confirm that a cause and effect relation exists between tides and upward gas-charged fluid migration and that a drop in hydrostatic pressure impacts the near-surface fluid dynamics by reducing the total stress and generating gas pressure pulses that lead to seepage.

**Conceptual models and implications**. Our data suggest that despite tidal height variations of less than 1 m, tidal cycles are able to cause a significant increase in the height of gas bubble plumes, reaching over 25 meters in continuous gas-equivalent height during the monitoring period (Figs. 5 and 6). Gas plumes are controlled by gas flow through porous sediments which may take place either by overcoming capillary resistance (capillary invasion), by opening existing fractures or by initiating new ones (fracture opening)[48].

**Capillary invasion**. A plausible explanation of gas flow through sediments is that the free gas pressure ($p_g$) at the location of the piezometer is under a pressure just below the capillary invasion pressure[48]. A decrease in pore water pressure ($p_w$) by less than 10 kPa (equivalent to 1 m) would then be sufficient for capillary pressure ($p_g - p_w$) to exceed the threshold value (right hand side of

Eq. (2)) corresponding to gas flow by capillary invasion (Eq. (2)).

$$p_g - p_w \geq \frac{2}{\sqrt{1 + \left(1 + \frac{d}{2r_g}\right)^2 - 1}} \frac{\gamma}{r_g} \approx 10 \frac{\gamma}{r_g} \qquad (2)$$

In Eq. (2), $\gamma$ is the interfacial tension between water and gas, $r_g$ is the grain radius and $d$ is the throat gap. By considering that the maximum $\gamma$ value between gas and water is equal to 72 mN/m[49] and for $r_g$ value taken as equivalent to 5 μm (mean grain size of the considered sediment), capillary invasion pressure is calculated to be 144 kPa. A decrease in pore pressure by 10 kPa, equivalent to tide fluctuations, may therefore correspond to 7% of the capillary pressure threshold. A small decrease in pore water pressure may thus suffice to cause gas capillary invasion in the upper sedimentary layers. However, this scenario is not compatible with field observations of massive gas hydrates[37] recovered in sediment cores near PZM1 site. A shallow front of free gas under the in-situ temperature and pressure conditions of PZM1 and PZM2 would be expected to form gas hydrates. For the observed thermal conditions and in situ thermal gradients (Supplementary Fig. 9) and salinity of 34 PSU, the base of the gas hydrate stability zone (GHSZ) is calculated to be at 145 m below the seabed at PZM1 and 170 m at PZM2. Thereby, free gas is expected to form gas hydrate above the GHSZ providing a reason to discard capillary invasion as a process controlling observed gas emissions.

**Fracture opening**. Terzariol and co-authors[50] defined a diagram relating the free gas circulation type to capillary pressure and in-situ effective stress. For the shallow in-situ effective stress conditions at site PZM1 (around 45 kPa at the tip of PZM1) and the calculated capillary pressure (144 kPa), the diagram indicates that fracture opening is the most probable mechanism explaining the observed free gas emissions during low tide cycles. This corroborates previous observations[48] showing that fracture opening is more favorable for gas circulation through fine-grained sediments and may occur when gas pressure exceeds the minimum principal stress ($\sigma_h$) according to the following equation:

$$p_g - \sigma_h \geq C_{LEFM} \cdot \frac{K_{IC}}{\sqrt{\pi a}} \qquad (3)$$

Where $K_{Ic}$ is fracture toughness, $a$ is the length of the fracture, and $C_{LEFM}$ is a coefficient that depends on the geometry, the ratio of horizontal to vertical stresses, and loading conditions[48]. Equation (3) is valid for linear elastic fracture mechanics (LEFM). For PZM1 site, none of the parameters of Eq. (3) are known to

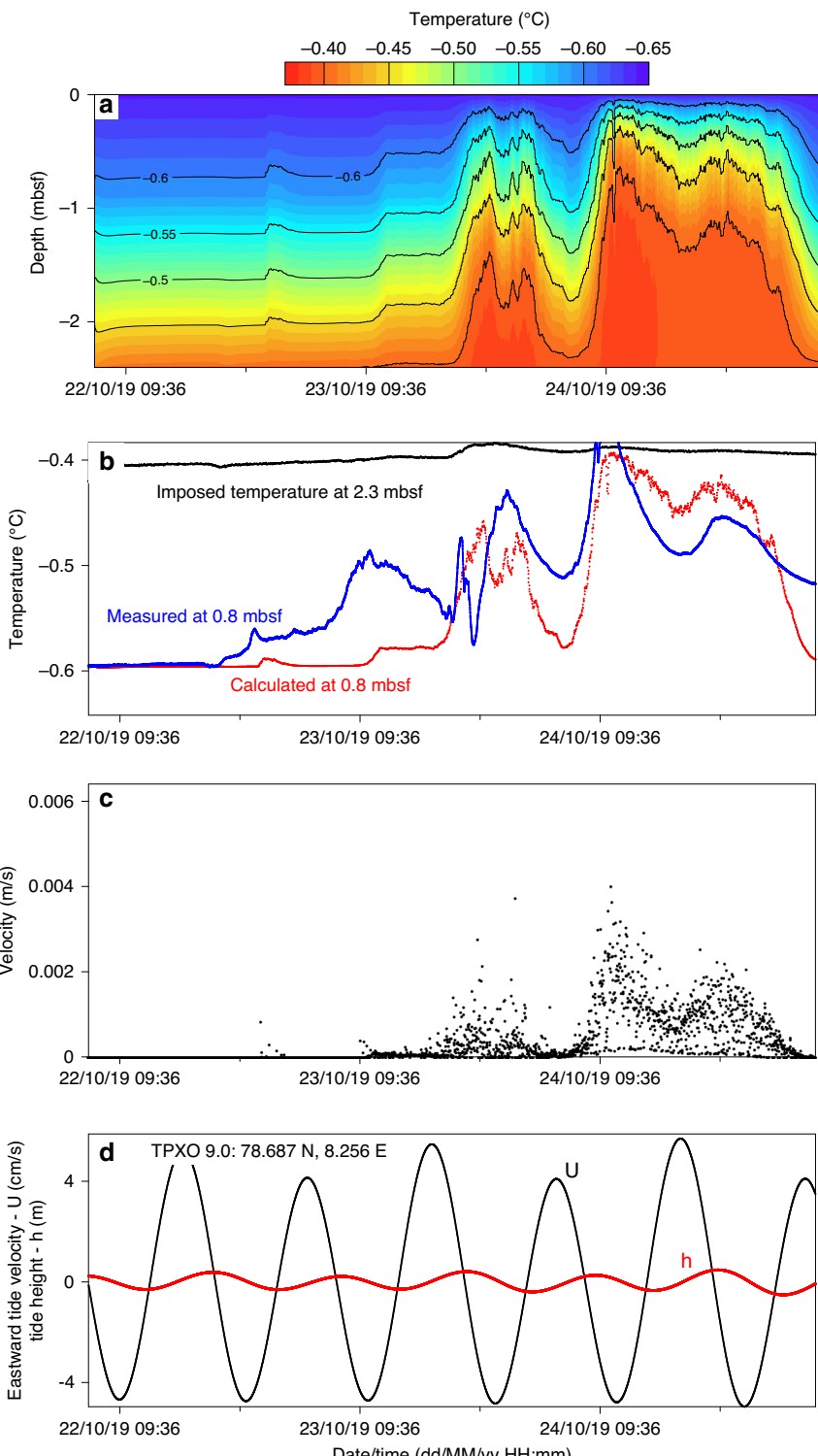

**Fig. 6 Thermal and hydraulic processes at PZM1. a** Predicted temperature field in the calculation domain. **b** The calculated temperature at 0.8 mbsf (red curve) is compared to the measured one (blue curve). **c** The upward fluid velocity used in the advection calculation. **d** The black curve indicates eastward tide velocity, while the red one is the tide height.

calculate the required $P_g$ to generate fracture opening. However, it is plausible that pre-existing fractures at the study area[15,23,51] get saturated by free gas from connected intermediate gas reservoirs or from the free gas front below the GHSZ; thus explaining the inferred free gas emissions (i.e., not visible in water column acoustic data but revealed by the piezometer pressure and temperature logs). Indeed, a reduction in hydrostatic pressure

affecting the gas volume and pressure ($P_g$) within the pre-existing fractures and gas reservoirs may cause $P_g$ to exceed the pressure threshold needed to generate fracture opening (Fig. 7). A similar process was proposed by Scandella and co-authors[40] to describe gas release and methane transport in lake sediments controlled by conduits dilation. They[40] show that a hydrostatic pressure deviation of 4 kPa is enough to reduce the effective stress below a

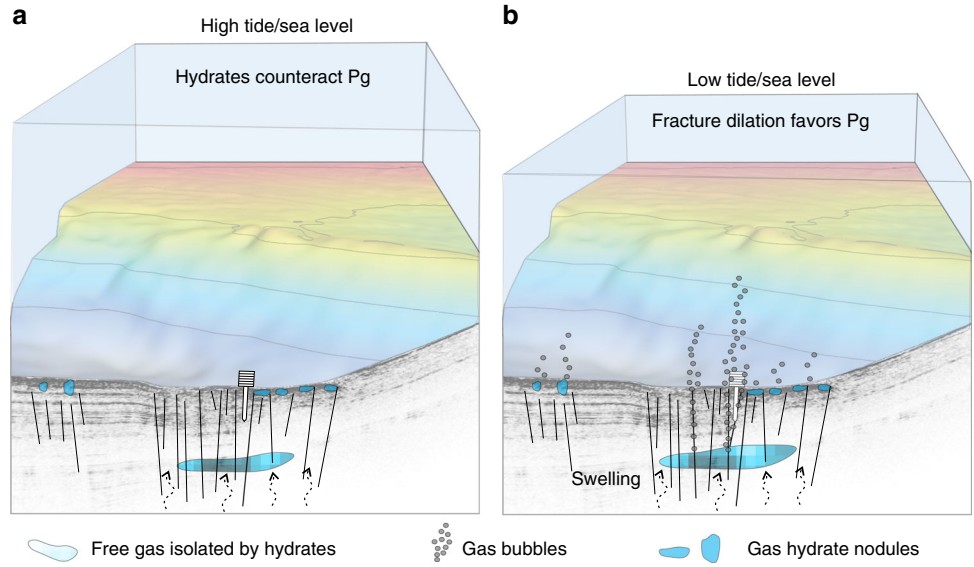

**Fig. 7 Conceptual model of gas emissions.** Schematic model showing how gas emissions may be affected by tides through fracture opening. **a** At high tides the system is under a balanced pressure field. **b** At low tides (<1 m water column height decrease) a subtle fracture dilation would shift the pressure field to favor gas advection and seepage. $P_g$ stands for free gas pressure.

pressure threshold corresponding to the tensile strength of the sediments and to generate important gas release.

We envision that at high tides the system is under a balanced pressure field where the minimum principal stress and the strength of patches of gas hydrates are enough to counteract the gas pressure in the system. A gas front reaching the near-surface from a deeper reservoir would either form or be slowed down by gas hydrates. At low tides, however, a subtle fracture dilation would shift the pressure equilibrium to favor gas advection and seepage (Fig. 7). These intermittent gas emissions due to tidal cycles are expected to prevent formation of massive and continuous gas hydrate layers near the seabed as it is shown in Fig. 7. Chaotic facies in the seismic data near the seabed are probably the expression of the occurrence of isolated free gas/gas hydrate nodules.

Data from the two investigated sites show a strong contrast between the pressure and temperature fluctuations with tides. While data from PZM1 suggest the occurrence of several events of bubbles rising through the water column during low tide cycles, data from PZM2 recorded small pore pressure fluctuations resulting from the seawater-free gas mixture fluid compressibility. Tides are expected to affect the hydrostatic pressure in the same manner at both sites. The contrast in collected pressure and temperature data may be thus related to differences in the distribution of fluids at a background versus a focused fluid flow site. It is plausible as well that a spatial variation in the tectonic stress field across the continental margin[52] has a local effect on the effective stress. Our results show, for the first time, that even a moderate sea-level rise (<1 m) may significantly impact gas bubble emissions at deep-water depths. On the other hand, expected degassing caused by warming ocean temperatures is expected to result in a positive climate feedback[53] and the question remains as to whether sea-level rise due to melting polar ice could partially counterbalance this anticipated phenomena.

Piezometer data from two sites offshore western Svalbard (Vestnesa Ridge) document the vulnerability of shallow gas accumulations to subtle changes in hydrostatic pressure. Pressure and temperature fluctuations are strikingly coincident with tidal cycles, with negative pressures corresponding to degassing during low tides. Although capillary invasion is a plausible mechanism

for explaining degassing in response to a decrease in hydrostatic pressure during low tides, such a model is unlikely in this deep-marine gas hydrate system. Rather, a fracture opening model seems ideal for explaining low tides induced degassing. Such a model reconciles observations of near-surface gas hydrates and inferred isolated free gas pockets. The joint analysis of temperature and pressure fluctuations point toward spatial variations in the amount and timing of gas advection towards the seafloor along the Vestnesa Ridge. The results illustrate the extra-sensitivity of the gas emission systems at deep-water sites to sea-level changes. Sea-level drops (low tides) seem to significantly affect near-surface free gas dynamics, even in deep-water conditions, causing important gas emissions. These results show that even a moderate sea-level rise (<1 m) may significantly reduce gas emissions and partially counterbalance future temperature effects on the global shallow marine gas systems.

Our results show that monitoring pore pressure in near-surface sediment is an approach that allows to constrain seafloor gas emissions and their governing processes beyond the limited capabilities of hydro-acoustic surveys. However, it is important to highlight the limitations of fixed-point observations with respect to the spatial coverage of a hydro-acoustic survey. Both methods are complementary and combining hydro-acoustic data and in-situ piezometer seems to be an effective approach to significantly reduce the uncertainties of emission rates due to temporal variability. We envision as next step the installation of long-term piezometer observatory offshore Svalbard to acquire long-time data series combined with recurrent hydro-acoustic surveys to further test our hypothesis and upscale our initial results for the broader Arctic. Such an approach is expected to improve predictive models of seabed gas emissions due to sea-level rise.

## Methods

**Piezometer**. In-situ pore pressure measurements are carried out using cable-deployed piezometer[54] equipped with a sediment lance carrying differential pore pressure and temperature sensors and ballasted with lead weights (up to 1000 kg, Fig. 8). The length of the lance is between 8.42 m (PZM1) and 9.92 m (PZM2) and is adapted to the stiffness of the sediment of the Vestnesa Ridge. The piezometer is connected to a sea-surface buoy by means of a synthetic rope.

Differential pore pressures (Supplementary Fig. 8) are measured relative to hydrostatic pressure at nine different ports on the 60 mm diameter lance using

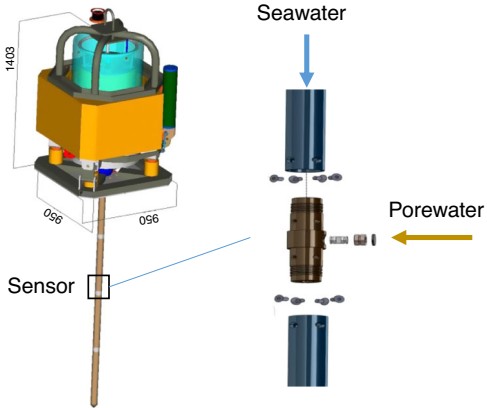

**Fig. 8 Piezometer.** Scheme of the piezometer equipped with differential pore pressure and temperature sensors and ballasted with lead weights. Measured negative pore pressures correspond to the case where the seawater hydrostatic pressure is higher than the sediment pore-water pressure.

specially adapted differential pressure gauges connected to the open seawater. The piezometer pore pressure and temperature sensors have an accuracy of ±0.5 kPa and 0.05 °C, respectively.

**Modeling: transient diffusion-advection heat transfer model.** To understand how pore-pressure variations and fluid advection may affect the temperature field, a 1D transient diffusion-advection heat transfer Eq. (4) is solved numerically using a finite difference method.

$$\frac{\partial T}{\partial t} = \frac{\partial}{\partial z}\left(k\frac{\partial T}{\partial z}\right) + v_z\frac{\partial T}{\partial z} \qquad (4)$$

In Eq. (4), $k$ signifies the thermal diffusivity of the material, $v_z$ the vertical velocity of the fluid, $T$ the temperature field, $t$ the simulation time and $z$ the depth below the seabed. To solve numerically the 1D diffusion-advection heat equation, a centered explicit finite difference discretization scheme is used.

## Data availability

The datasets generated and analyzed during the current study are available from the corresponding author on reasonable request.

## Code availability

Computer code used to generate results that are reported in Fig. 6 is available from the corresponding authors upon reasonable request.

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

## Acknowledgements

This study was conducted in the framework of the SEAMSTRESS project and its grants from the Tromsø Research Foundation (TFS) and the Research Council of Norway (grant 287865). The work is also supported by the Research Council of Norway through its Centres of Excellence funding scheme, project 223259 and by the ISblue project, Interdisciplinary graduate school for the blue planet (ANR-17-EURE-0015) and co-funded by a grant from the French government under the program "Investissements d'Avenir". We are greatly thankful to the crew onboard R/V Kronprins Haakon for their support during conduction of the piezometer experiment. Special thanks to Mickael Roudaut and Pierre Guyavarch from Ifremer and Jan Bremnes former IMR for ensuring that deployment of the piezometer was technically possible. The corresponding author thanks insightful discussion with Pascale Lherminier, Stephan Ker, Vincent Riboulot, Laurent Berger, and Carla Scalabrin and acknowledges Alison Chalm for language editing.

## Author contributions

N.S. led the piezometer experiment, conducted data acquisition and analyses, and wrote the paper. A.P.-F. conceived and designed the Seamstress research project and contributed to data analyses. S.V. contributed to data acquisition and analyses. S.B. contributed to data analyses. J.K. led the CAGE 19-3 oceanographic cruise. A.P.-F., S.V., S.B., and J.K. helped write the paper.

## Competing interests

The authors declare no competing interests.
