## [Peer Review File · Nature Communications]

REVIEWER COMMENTS

Reviewer #1 (Remarks to the Author):

The authors present new in-situ temperature and pressure monitoring data for two sites at the Vestnesa Ridge. Even though their study area was previously classified as inactive in term of gas and fluid flow, the author find P/T-signatures indicating the presence of gas in shallow sediments and active gas flow during low-tide periods. It was previously shown for active gas seepage sites that tides affect seabed pressure and gas flow. The authors argue that these effects can also occur at sites that are apparently inactive. As far as I can tell the data and the evaluation are sound and yield interesting hypotheses. However, I am not an expert in the interpretation of piezometer profiles and hope that other reviewers are involved in the review process to cover this important aspect and validate the uniqueness of the data interpretation.

The paper would be much stronger if the conclusions derived by the evaluation of in-situ temperature and pressure monitoring data would be supported by additional independent data taken at the two monitoring sites. As far as I know, hydro-acoustic methods should be sensitive enough to detect gas plumes with a height of up to 25 m predicted by the P/T data evaluation. It is thus somewhat surprising that this gas plume signal was not detected during the hydro-acoustic surveys. The authors should better explain why this is not the case, i.e. why the gas plume signal was not detected during hydro-acoustic surveys. Moreover, shallow seismic data (e.g. Parasound are equivalent methods) could be used to detect free gas in the upper 10 m of the sediment column and confirm the interpretation of the P/T data. The authors should add such data to the paper if they are available. The authors do not report on sediment core data taken at their monitoring sites. These data would be very important and could be used to corroborate the hypotheses derived from the P/T-monitoring data since the presence of a free gas phase should induce gas hydrate formation and anaerobic methane oxidation (AOM) that can be detected by characteristic porewater signatures (decrease in dissolved sulfate, increase in dissolved sulfide and depletion in dissolved chloride due to hydrate dissociation upon core retrieval). Hence, I would ask the authors to include such data if they are available.

Furthermore, a few minor points should be addressed by the authors:

Hours or days rather than seconds should be used for the time axis in Figs 2 and 3. The arrow in Fig. 2a seems to point in the wrong direction.

The temperature data in Fig.2 are interpreted assuming that the bottom water temperature (BWT) as constant over the monitoring period. However, it may be possible that tidal currents induced BWT changes over this period. The authors should address/exclude this possibility.

Please replace "finite deference" method by "finite difference" method in section "Modeling. Transient diffusion-advection heat transfer model".

Reviewer #2 (Remarks to the Author):

Temperature and pore-pressure were measured in the upper 10 m of sediment in an area of known seafloor methane seepage and gas hydrates, but away from known gas flares. Sub-seabed upward fluid flow and tidally modulated gas release from the seabed were observed. Two key implications are discussed: (1) present day estimates of seabed methane release may be too low for this and similar sites, as hydro-acoustic flare mapping in the area had not identified intermittent tidally-modulated gas escape into the water column, and (2) future sea-level rise may counteract the predicted future increase in methane release from such systems due to warming-induced hydrate destabilisation to a larger degree than expected, given the significant impact of the relatively small tidally-driven changes in pore water pressure.

Tidally modulated methane release from the seabed and its implications for present and future emission estimates, including in the Arctic, has been previously discussed in the literature, but use of piezometers to directly observe the sub-surface pressure and gas migration in these systems is novel. In particular, the insight into mechanisms for how small changes in pressure affect gas release from the seabed will be of

interest to the community and wider field because of their direct relevance to understanding the future impacts of sea-level rise on these systems.

While I find the results and analysis of the study appropriate for publication in Nature Communications, I find the overall narrative challenging to follow. It therefore fails convey a compelling message in its current form. Greater attention to detail is recommended throughout to avoid distracting readers from the ideas presented. Careful re-evaluation of the text, not limited to the suggestions below, would make the objectives of the work, key observations, and their important implications more accessible to a broader readership. The existing sub-headings provide a good framework within which to achieve this.

Specific and general comments:

General: review would have been made easier by the inclusion of line or paragraph numbers. I hope that my reference to paragraph numbers within named sections (as P#) is clear.

Abstract: The constraints of the word limit are evident from first reading, while the main ideas are less clear. The use of shorter sentences might help avoid confusion. It is difficult to separate the two key results: gas release at previously unmapped sites, and tidal effects – and their opposing suggested impacts to methane release of “underestimated” (too low) and “can reduce the volume of” (too high).

Introduction

P1: there is an important distinction between methane emissions from the seabed into ocean bottom waters, and emission of methane from the ocean into the atmosphere where it acts as a greenhouse gas. Please ensure wording does not imply that all methane escaping the seabed contributes to the greenhouse effect in the atmosphere.

P5: “visible effect”: consider rewording. Observable/Measurable? Important? (to what? – local/global emissions?)

P6: “We suggest that tides can be used as a proxy to predict and quantify variability in the amount of gas released on a daily basis.” – this is interesting and important, but somewhat misleading in that the methodology for such an approach is not presented, only support for it being possible. Consider rewording to more clearly distinguish what is done within this paper from the broader implications/next steps. Elsewhere, consider further discussing how the new observations from the piezometers will allow this to be achieved in the future – how they advance the field towards this goal beyond what was already understood from previous studies showing the impact of tidal pressure changes on gas release by e.g. bubble flare timeseries. A revisit to the use of tides as a proxy is missing from the final sections.

P6: “the most known active gas emission site in the Arctic”: subjective. Suggest: ‘among the best studied’, or similar, as appropriate.

Distribution of seafloor seepage of western Svalbard

P1: “very discrete locations”: ‘very’ is unnecessary/unclear. If it is intended to imply a meaning different from ‘discrete locations’ please clarify.

Figure 1: “surveyed gas plumes during yearly expeditions to the area” over how many years (assuming that the number of years is equal to the number of surveys?)?

Results

General: it would be easier for readers to follow the discussion through the text if shortened names for the two deployments were introduced. Maybe PZM1 and PZM2? Or even Site 1 and Site 2.

P1: why were non-active sites (no previously observed flares) chosen?

General: Many words are used which unnecessarily repeat the information provided in Table 1, e.g. the first sentences of both paragraph 2 and paragraph 3; restating the name of the vessel from which the instruments were deployed more than once does not help a reader pick out the important results from the data shown.

Figure 2: remove @ signs in legend. Is the 2nd decimal place in the sensor depth (to cm precision)

appropriate (consider this also throughout the text elsewhere)? Consider changing the time units on the x axis to be more appropriate for their magnitude and inserting additional axis ticks for easier interpretation. Figure 2 (caption): "Negative pore pressure is an indicator of upward free gas migration". This message is not clearly conveyed by the text in the preceding paragraph (which states "deviation towards warmer temperatures indicative of upward fluid flow" and "pressure fluctuation reflects the effect of changes in hydrostatic pressure on compressible fluid (ie gas)". The long figure captions throughout combine both a detailed description of the data shown and their key implications, which are also presented in the text. Readability of this approach would be improved by being more concise in the figure caption text and ensuring that any synthesis of the data given in the figure caption is not missing from the main text. For example: The first sentence "Sensor depths are between 0.79 mbsf (grey curve) and 7.94 mbsf (brown curve)." is repeating information already available in the figure legend, but it might be helpful to instead state that "each coloured line represents a different sensor depth". The following two sentences of the caption provide a useful explanation for the initial portion of the signal, but at a level of detail perhaps more appropriate for the main text than a caption, the information needed to understand the figure requires only a few words here. The lengthy sentence "Temperatures at the deepest eight sensors are almost constant throughout the monitoring period whilst the uppermost sensor recorded important positive temperature fluctuations indicative of upward fluid flow." repeats verbatim text in the paragraph above. Suggest removing this here. The subsequent two sentences are short and clear, communicate what is needed for a reader to be able to infer that idea from the data shown. Consider the above comments also with reference to Figure 3.

Seawater-Sediment Interactions:

General: section presents interesting discussion but is difficult to parse. Suggest splitting first paragraph into multiple, perhaps separating the general background of piezometer data from the discussion of the results of PZM2. Consider that the target readers (of Nature Communications) may not be familiar with PZM data interpretation and include a few more words from previous studies to allow the logic to be followed rather than assuming the reader will either be aware of the cited literature or take (have) the time to read these papers. Why does "The difference in the compressibility of seawater saturating the piezometer rod and the sediment pore fluid makes the response of the four shallowest differential pressure sensors of KH-05-PZM2 sensitive to tidal cycles." Is this related to their depth (so true for every PZM deployment in general), is it specific to the water depth of deployment? A few more words are needed here.

Penultimate sentence: Why does this "suggest upward transport of free gas through the sediments". Add a short sentence after this to help readers connect the dots.

Final sentence: Suggest removing.

Rising velocity of gas bubbles

P1: The precise times of the subset of data shown are not important for the reader's understanding of what is discussed here. The first two sentences could be omitted/incorporated into the third with reference to the figure, and this would make the key information presented in this section more accessible.

Figure 4: Is 2nd decimal place precision justified by the measurements/calculation? Remove the '00' seconds from the x axis. Suggest colouring the different sensor outputs by depth in (a) to create consistency across the figures. Consider comments from Figure 1 in making the caption more concise (e.g., restating the time interval which is clearly shown on the x axis does not help with figure interpretation). Suggest leaving off the final sentence, as this idea is better communicated in the paragraph below.

Paragraph following Equation 1: Final sentence provides information which should have been included in the results section when it was first stated that high temperatures indicate upward fluid flow. Here, a sentence explaining why it makes sense for upward fluid flow and gas escape to the seafloor to co-occur would be more appropriate.

Figure 5: please refer to comments on preceding figures related to x-axis label format and including more appropriate sub-divisions. Remove unnecessary white space at beginning and end of timeseries. Rephrase "correctly correlate"; use of 'correctly' is inappropriate. Consider if 'correlate' (with statistical implications) conveys the desired message here, which seems to be 'occur at approximately the same time as'?

Tides as a forcing mechanism

P1, 1st sentence: Referring back to "data in Figure" is a distraction from the key idea of this sentence, it

sends readers scrolling (or flipping) back up to the figure and which breaks the thread of the narrative. It would be clearer to start by reminding readers what is shown in figure 2 (which is done with the text at the end of this sentence), before building on that with the suggested implication.

P1 "imposed temperature measured at the base of the calculation": A measurement was not made in the calculation. Do you mean "the base of the model domain"?

P2 "confirming that vertical fluid advection is most likely the source of". The model shows that it provides an explanation for, which fits most of the observations, but no other 'likely' explanations have been explored or shown to be less likely than that presented. Either add to the discussion to better support this statement or reword.

P3, first sentence: the ability to compare the duration of pulses does not depend on the stated assumption. I understand this sentence to indicate that the interest in making this comparison, or it's usefulness to furthering the discussion, depends on this assumption(?) It would be much more instructive to explain why this is the case.

P3, "fit reasonably well with": be more specific, maybe 'correspond in time to'? Is there a lag? How good is "reasonably"?

P3, "This correlation suggest that tides are impacting the near-surface fluid dynamics, working as a forcing mechanism of gas pressure pulses and active seepage." Please add some words to help readers make the step from considering tidal velocity (mentioned in the previous sentence) to a pressure-mediated (?) control on subsurface fluid flow. It seems more intuitive that the comparison would be between tidal height (also shown in the figure) here, but this is not mentioned in the text. Also consider: the use of the word "correlation" implies a statistical comparison; "suggest" should be "suggests".

Figure 6: please consider comments for other figure captions and time axis labels. There is no need for the vertical grey lines to extend across sub-figures.

Discussion

Capillary invasion:

General: this section is missing a one-sentence explanation of what capillary invasion is. Or "capillary invasion" as referred to in the final sentence (what is meant to be implied by the quotation marks here, when they are not used elsewhere?).

P1, "preferential paths through sediment layers": please clarify what is meant by this in the context used.

P2, 2nd sentence: where have these cited numbers come from? References? Measurements of the local average grain size during this study? And, subsequently, the a geothermal gradient of 87 °C/km is used – is this based on the current study, or missing a citation?

P2, "However, this scenario is not compatible with field observations of massive gas hydrates in sediment cores recovered near KH-01-PZM1 site.": why not? Add this information. This paragraph seems to end by stating that the simple fact that both PZM were deployed within the GHSZ means that the mechanism discussed is not possible. If this is the case, why bother presenting it in such detail? If this is not the case, please clarify.

Include the calculated base of the GHSZ for PZM2, for completeness. Readers less familiar with GHSZ work would benefit from the addition of the words "above which free gas is expected to form/be trapped in gas hydrates", or similar.

Fracture Opening

P1: This is difficult to follow. Please define terms more clearly, instead of as introducing in a series of bracketed insertions.

P2 "None of the parameters of equation (3) are known...", In general, or for the site(s) in question? Please clarify.

P3. This paragraph is very clear. Minor comment: are gas emissions "continuous" if gas only escapes during part of the tidal cycle?

Summary

General: in noting that PZM appear to be able detect seafloor gas seepage which was missed by hydroacoustic surveys, it seems appropriate to consider that the fixed-point observation of a PZM has several limitations compared to the spatial coverage of a hydroacoustic survey. Above, it would be interesting to have the 'no flares observed' at the PZM sites better parameterised: how many times have

they been surveyed over how many surveys? Was the site being observed by hydroacoustics during the gas escape observed by the PZM?

General: Thinking more broadly, would it be possible/appropriate to apply the new information provided by the PZMs to the gas emission estimates for the Vestnesa Ridge or broader western Svalbard area to provide an example of how this information can be used? I expect the short answer to this is no, but this final section is missing an element of looking forward towards this type of goal to add perspective to the "significantly affect" and "significantly reduce" of the final two sentences. This comment links back to that above related to the final statement of the introduction section.

Methods: check consistency with use of present and future verb tense.

The authors thank the referees for their very constructive comments. In the following, we give a point-to-point reply (blue) to the referee comments

Reviewer #1	
Comments	Reply
R1 - The paper would be much stronger if the conclusions derived by the evaluation of in-situ temperature and pressure monitoring data would be supported by additional independent data taken at the two monitoring sites.	We included additional data in a supplementary material file. These data include sub-bottom profiles, bathymetry, P wave velocity data and images of gas hydrate nodules from a gravity core.
R2 - As far as I know, hydro-acoustic methods should be sensitive enough to detect gas plumes with a height of up to 25 m predicted by the P/T data evaluation. It is thus somewhat surprising that this gas plume signal was not detected during the hydro-acoustic surveys. The authors should better explain why this is not the case, i.e. why the gas plume signal was not detected during hydro-acoustic surveys.	Timing of the survey is probably an important factor in determining why the seepage at PZM1 site was not spotted in sonar profiles. Our data show that gas emissions are strongly influenced by tide cycles but they are also dependent on the dynamics of subseabed gas accumulation. These near-surface gas accumulations experience pulses of pressure build-up and depletion and emissions seem to occur only when a threshold pressure is reached (equation 3) making the system intermittent. It is reasonable to assume that for the seepage to be seen as a gas plume in sonar data, the ship would have to sail over the seepage at the exact time of emission. We have included more reasoning on this in the main text. In the supplementary material, we have added the dates of all the hydro-acoustic data recovered from the PZM1 study site. None of these data indicate the presence of gas plumes in the water column confirming that sporadic surveying is not sufficient to characterize such a dynamic system. We argue that these sporadic observations using hydro-acoustic surveys can only provide a partial image of gas emissions from an active but intermittent seepage system.
R3 - Moreover, shallow seismic data (e.g. Parasound are equivalent methods) could be used to detect free gas in the upper 10 m of the sediment column and confirm the interpretation of the P/T data. The authors should add such data to the paper if they are available.	Sub-bottom and chirp profiles reveal widespread occurrences of transparent seismic facies typically indicating a vertical gas migration system (acoustic masking, discontinuities, high amplitudes). We have added interpretation of these characteristics of vertical plumbing systems as supplementary material. We have included a sub-bottom profile line that runs along a north-south seafloor cavity with sub-depressions associated with vertical fluid

	migration pathways (PZM1 is at the flank of the northernmost depression).
R4 - The authors do not report on sediment core data taken at their monitoring sites. These data would be very important and could be used to corroborate the hypotheses derived from the P/T-monitoring data since the presence of a free gas phase should induce gas hydrate formation and anaerobic methane oxidation (AOM) that can be detected by characteristic porewater signatures (decrease in dissolved sulfate, increase in dissolved sulfide and depletion in dissolved chloride due to hydrate dissociation upon core retrieval). Hence, I would ask the authors to include such data if they are available.	We have added to the supplementary material pictures of gas hydrate nodules retrieved within 0.7-1.2 mbsf on gravity core CAGE17-5_1401 located ca. 1.5 km from PZM1 south along the major seafloor cavity. This is a direct indication of gas accumulation near the seabed. We have also added to the supplementary material P wave velocity data obtained from gravity core recovered in 2019 near PZM1 showing very low values possibly related to the presence of in-situ dissolved free gas. The sub-bottom profile added to the supplementary material (see previous comment) shows that P/T data from PZM1 is from a setting similar to that of the gas hydrate core to its south (i.e., characterized by acoustic masking and vertical discontinuities that correspond to the upper part of gas chimneys, often better depicted in seismic data). Supplementary figure 2, indicates that the recovered cores near PZM1 (GC_01 and GPC_01) are from area characterized by layered and undisturbed reflectors. Available dissolved sulfate profiles from these cores do not show any traces of AOM and therefore we do not think that pore water data provide new conclusive elements to our discussion.
R5 - Hours or days rather than seconds should be used for the time axis in Figs 2 and 3.	Considered and revised
R6 - The arrow in Fig. 2a seems to point in the wrong direction.	Actually, the arrow is correct - The shallowest sensor is characterized by the lowest temperature. Nevertheless, we have removed the arrow to avoid confusion.
R7 - The temperature data in Fig.2 are interpreted assuming that the bottom water temperature (BWT) as constant over the monitoring period. However, it may be possible that tidal currents induced BWT changes over this period. The authors should address/exclude this possibility.	We added the following sentence : “Although tidal currents may induce seabed temperature fluctuation, temperatures from the upper sensor of PZM2 remained almost constant during the 4-day monitoring period indicating that the impact of currents on the studied process was negligible.”
R8 - Please replace “finite deference” method by “finite difference” method in section “Modeling. Transient diffusion-advection heat transfer model”.	Corrected
Reviewer #2	
Comments	Reply

Abstract	
The constraints of the word limit are evident from first reading, while the main ideas are less clear. The use of shorter sentences might help avoid confusion. It is difficult to separate the two key results: gas release at previously unmapped sites, and tidal effects – and their opposing suggested impacts to methane release of “underestimated” (too low) and “can reduce the volume of” (too high).	Considered and rephrased with shorter sentences.
Introduction	
P1: there is an important distinction between methane emissions from the seabed into ocean bottom waters, and emission of methane from the ocean into the atmosphere where it acts as a greenhouse gas. Please ensure wording does not imply that all methane escaping the seabed contributes to the greenhouse effect in the atmosphere.	Rephrased: “Given the impact of methane as a greenhouse gas, the dynamic of oceanic methane emissions, which could potentially reach the atmosphere, introduces a non-negligible doubt on the global budget of atmospheric methane.” The last part of the sentence concerning “climate change projections” has been removed.
P5: “visible effect”: consider rewording. Observable/Measurable? Important? (to what? – local/global emissions?)	Rephrased: may have “a global impact” on the intensity of deep-sea gas emissions
P6: “We suggest that tides can be used as a proxy to predict and quantify variability in the amount of gas released on a daily basis.” – this is interesting and important, but somewhat misleading in that the methodology for such an approach is not presented, only support for it being possible. Consider rewording to more clearly distinguish what is done within this paper from the broader implications/next steps.	The last sentence of P6 explains the factual methodology: “We carried out in-situ sediment pore-pressure and temperature measurements... We aim at better characterizing the short-term periodicity of deep-marine seepage and the effect of tides on the pressure field that controls the emissions.” In the new version of the manuscript (in “Seawater-Sediment Interactions”) we detailed the physical process and methodology needed to use piezometer data as an indicator of gas emissions and gas accumulation: “measured differential pore pressure during a change in sea level becomes an indicator of the type of fluids saturating the sediment as described below: - For seawater fully saturating sediment pores, a change in the height of the seawater column (i.e. tides) is expected to affect both sides

Elsewhere, consider further discussing how the new observations from the piezometers will allow this to be achieved in the future – how they advance the field towards this goal beyond what was already understood from previous studies showing the impact of tidal pressure changes on gas release by e.g. bubble flare timeseries. A revisit to the use of tides as a proxy is missing from the final sections.	of the differential sensor equally without any change in the recorded pore pressure. - In the presence of free gas within the sediment pores, a change in the height of the seawater column is likely to affect the measured differential pore pressure because of the high compressibility of the free gas with respect to the seawater. A drop in hydrostatic pressure related to tides reduces the total stress and thus causes free gas expansion and exsolution thereby increasing the compressibility of the sediment pore fluid and disturbing the measured differential pore pressure. - Free gas emissions can be also captured by differential pore pressure gauges since the sensors measure the difference between hydrostatic pressure related to seawater column height (water depth) and sediment pore pressure. A negative differential pore pressure (pressure lower than the hydrostatic pressure) is therefore a strong indicator of the presence of low-density fluid saturating the sediment pores and migrating into the water column." We have added a paragraph describing the way we want to move forward on this research subject to the last section.
P6: "the most known active gas emission site in the Arctic": subjective. Suggest: 'among the best studied', or similar, as appropriate.	Rephrased
Distribution of seafloor seepage of western Svalbard	
P1: "very discrete locations": 'very' is unnecessary/unclear. If it is intended to imply a meaning different from 'discrete locations' please clarify.	Modified as suggested. It is indeed about "discrete locations"
Figure 1: "surveyed gas plumes during yearly expeditions to the area" over how many years (assuming that the number of years is equal to the number of surveys?)?	Details are now provided in the supplementary material
Results	
General: it would be easier for readers to follow the discussion through the text if shortened names for the two deployments were introduced. Maybe PZM1 and PZM2? Or even Site 1 and Site 2.	Names have been changed to PZM1 and PZM2

P1: why were non-active sites (no previously observed flares) chosen?	Added the following sentence: “We explicitly selected the sites to avoid active geological structures while characterizing the hydraulic and thermal properties of the near-surface sediment.”
General: Many words are used which unnecessarily repeat the information provided in Table 1, e.g. the first sentences of both paragraph 2 and paragraph 3; restating the name of the vessel from which the instruments were deployed more than once does not help a reader pick out the important results from the data shown.	The text has been simplified by removing repetitions.
Figure 2: remove @ signs in legend. Is the 2nd decimal place in the sensor depth (to cm precision) appropriate (consider this also throughout the text elsewhere)?	Considered and revised
Consider changing the time units on the x axis to be more appropriate for their magnitude and inserting additional axis ticks for easier interpretation.	Considered and revised
Figure 2 (caption): “Negative pore pressure is an indicator of upward free gas migration”. This message is not clearly conveyed by the text in the preceding paragraph (which states “deviation towards warmer temperatures indicative of upward fluid flow” and “pressure fluctuation reflects the effect of changes in hydrostatic pressure on compressible fluid (ie gas)”.	Considered. This is now discussed in more detail in the “Seawater-Sediment Interactions” paragraph.
The long figure captions throughout combine both a detailed description of the data shown and their key implications, which are also presented in the text. Readability of this approach would be improved by being more concise in the figure caption text and ensuring that any synthesis of the data given in the figure caption is not missing from the main text. For example: The first sentence “Sensor depths are between 0.79 mbsf (grey curve) and 7.94 mbsf (brown curve).” is repeating information already available in the figure legend, but it might be helpful to instead state that “each coloured line represents a different sensor depth”.	We intended to introduce in the manuscript self-explanatory figure captions. The reviewer’s comment is also relevant. We followed the reviewer’s comment and simplified the legend captions in the manuscript.
The following two sentences of the caption provide a useful explanation for the initial portion of the signal, but at a level of detail perhaps more appropriate for the main text than a caption, the information needed to understand the figure requires only a few words here. The	Considered and revised

lengthy sentence “Temperatures at the deepest eight sensors are almost constant throughout the monitoring period whilst the uppermost sensor recorded important positive temperature fluctuations indicative of upward fluid flow.” repeats verbatim text in the paragraph above. Suggest removing this here. The subsequent two sentences are short and clear, communicate what is needed for a reader to be able to infer that idea from the data shown.	
Consider the above comments also with reference to Figure 3.	Done
Seawater-Sediment Interactions	
General: section presents interesting discussion but is difficult to parse. Suggest splitting first paragraph into multiple, perhaps separating the general background of piezometer data from the discussion of the results of PZM2. Consider that the target readers (of Nature Communications) may not be familiar with PZM data interpretation and include a few more words from previous studies to allow the logic to be followed rather than assuming the reader will either be aware of the cited literature or take (have) the time to read these papers.	Considered. We have suggested a new general introduction explaining the way the piezometer data are used in terms of gas occurrence and gas emissions in more detail. We have added a new paragraph entitled “Evidence for the occurrence of free gas”
Why does “The difference in the compressibility of seawater saturating the piezometer rod and the sediment pore fluid makes the response of the four shallowest differential pressure sensors of KH-05-PZM2 sensitive to tidal cycles.” Is this related to their depth (so true for every PZM deployment in general), is it specific to the water depth of deployment? A few more words are needed here.	The presence of gas with high compressibility in the upper few meters makes the response of the four shallowest differential pressure sensors of PZM2 sensitive to tidal cycles. Rephrased “The four shallowest differential pressure sensors of PZM2 showed a response to tidal cycles indicating the presence of sediment gas-charged fluid.”
Penultimate sentence: Why does this “suggest upward transport of free gas through the sediments”. Add a short sentence after this to help readers connect the dots.	We have added this sentence for clarification: “Temperature peaks above the measured seawater temperature (i.e., -0.64 °C) in the upper sensor indicate upward fluid flow (Figure 2a).”
Final sentence: Suggest removing.	Considered
Rising velocity of gas bubbles	
P1: The precise times of the subset of data shown are not important for the reader’s understanding of what is discussed here. The first two sentences could be omitted/incorporated into the third with reference to the figure, and this would make the	Considered and revised

key information presented in this section more accessible.	
Figure 4: Is 2nd decimal place precision justified by the measurements/calculation? Remove the '00' seconds from the x axis. Suggest colouring the different sensor outputs by depth in (a) to create consistency across the figures.	Considered and revised
Consider comments from Figure 1 in making the caption more concise (e.g., restating the time interval which is clearly shown on the x axis does not help with figure interpretation). Suggest leaving off the final sentence, as this idea is better communicated in the paragraph below.	Considered and revised
Paragraph following Equation 1: Final sentence provides information which should have been included in the results section when it was first stated that high temperatures indicate upward fluid flow. Here, a sentence explaining why it makes sense for upward fluid flow and gas escape to the seafloor to co-occur would be more appropriate.	Considered and revised This is now the aim of the next paragraph
Figure 5: please refer to comments on preceding figures related to x-axis label format and including more appropriate sub-divisions. Remove unnecessary white space at beginning and end of timeseries. Rephrase "correctly correlate"; use of 'correctly' is inappropriate. Consider if 'correlate' (with statistical implications) conveys the desired message here, which seems to be 'occur at approximately the same time as'?	Considered and revised We replaced "correctly correlate" by "coincides"
Tides as a forcing mechanism	
P1, 1st sentence: Referring back to "data in Figure" is a distraction from the key idea of this sentence, it sends readers scrolling (or flipping) back up to the figure and which breaks the thread of the narrative. It would be clearer to start by reminding readers what is shown in figure 2 (which is done with the text at the end of this sentence), before building on that with the suggested implication.	Considered and revised "Upward gas-charged fluid migration was suspected to be responsible for the recorded negative differential pressure cycles and temperature fluctuations measured by the upper sensor of PZM1."
P1 "imposed temperature measured at the base of the calculation": A measurement was not made in the calculation. Do you mean "the base of the model domain"?	Revised "imposed temperature at the base of the model domain equivalent to the measured temperature"
P2 "confirming that vertical fluid advection is most likely the source of". The model shows that it provides an explanation for, which fits most of the observations, but no other 'likely' explanations have been explored or shown to be	Reworded "the four temperature peaks were reproduced indicating that vertical fluid advection may explain the measured temperature fluctuations."

less likely than that presented. Either add to the discussion to better support this statement or reword.	
P3, first sentence: the ability to compare the duration of pulses does not depend on the stated assumption. I understand this sentence to indicate that the interest in making this comparison, or it's usefulness to furthering the discussion, depends on this assumption(?) It would be much more instructive to explain why this is the case.	Rephrased "Moreover, the duration of each thermal pulse can be compared to the period of tide cycles." The explanation about the link between tides and thermal pulses is developed at the end of P3.
P3, "fit reasonably well with": be more specific, maybe 'correspond in time to'? Is there a lag? How good is "reasonably"?	Replaced by "correspond in time to"
P3, "This correlation suggest that tides are impacting the near-surface fluid dynamics, working as a forcing mechanism of gas pressure pulses and active seepage." Please add some words to help readers make the step from considering tidal velocity (mentioned in the previous sentence) to a pressure-mediated (?) control on subsurface fluid flow. It seems more intuitive that the comparison would be between tidal height (also shown in the figure) here, but this is not mentioned in the text.	We added the following sentence: "These results confirm that a cause and effect relation exists between tides and upward gas-charged fluid migration and that a drop in hydrostatic pressure impacts the near-surface fluid dynamics by reducing the total stress and generating gas pressure pulses that lead to seepage."
Also consider: the use of the word "correlation" implies a statistical comparison; "suggest" should be "suggests".	Considered and revised
Figure 6: please consider comments for other figure captions and time axis labels. There is no need for the vertical grey lines to extend across sub-figures.	Considered and revised
Discussion	
Capillary invasion General: this section is missing a one-sentence explanation of what capillary invasion is. Or "capillary invasion" as referred to in the final sentence (what is meant to be implied by the quotation marks here, when they are not used elsewhere?).	We have added the following definition: "Gas plumes are controlled by gas flow through porous sediments which may take place either by overcoming capillary resistance (capillary invasion), by opening existing fractures or by initiating new ones (fracture opening)." quotation marks removed
P1, "preferential paths through sediment layers": please clarify what is meant by this in the context used.	Rephrased. "corresponding to gas flow by capillary invasion"
P2, 2nd sentence: where have these cited numbers come from? References? Measurements of the local average grain size during this study? And, subsequently, the a	Added a reference for the interfacial tension. Mean grain size is of the considered sediment

geothermal gradient of 87 °C/km is used – is this based on the current study, or missing a citation?	Thermal gradient is obtained from the piezometer data (now added to supplementary material)
P2, “However, this scenario is not compatible with field observations of massive gas hydrates in sediment cores recovered near KH-01-PZM1 site.”: why not? Add this information. This paragraph seems to end by stating that the simple fact that both PZM were deployed within the GHSZ means that the mechanism discussed is not possible. If this is the case, why bother presenting it in such detail? If this is not the case, please clarify.	This is now explained in the following sentence “Thereby, free gas is expected to form gas hydrate above the GHSZ providing a reason to discard capillary invasion as a process controlling observed gas emissions.” We discuss here the validity of an alternative scenario. We cannot say that fracture opening is the unique valid mechanism without evaluating the capillary invasion process.
Include the calculated base of the GHSZ for PZM2, for completeness. Readers less familiar with GHSZ work would benefit from the addition of the words “above which free gas is expected to form/be trapped in gas hydrates”, or similar.	Done Added “Thereby, free gas is expected to form gas hydrate above the GHSZ providing a reason to discard capillary invasion as a process controlling observed gas emissions.”
Fracture Opening P1: This is difficult to follow. Please define terms more clearly, instead of as introducing in a series of bracketed insertions.	Sentence simplified
Fracture Opening P2 “None of the parameters of equation (3) are known...”, In general, or for the site(s) in question? Please clarify.	For site PZM1, this is now clearly indicated
Fracture Opening P3. This paragraph is very clear. Minor comment: are gas emissions “continuous” if gas only escapes during part of the tidal cycle?	Considered Continuous replaced by intermittent
Summary	
General: in noting that PZM appear to be able detect seafloor gas seepage which was missed by hydroacoustic surveys, it seems appropriate to consider that the fixed-point observation of a PZM has several limitations compared to the spatial coverage of a hydroacoustic survey.	Considered. We have added the following sentence: “However, it is important to highlight the limitations of fixed-point observations with respect to the spatial coverage of a hydroacoustic survey. Both methods are complementary and their combination can be used to characterize gas emissions in a wide range of distances and timescales.”
Above, it would be interesting to have the ‘no flares observed’ at the PZM sites better parameterised: how many times have they been surveyed over how many surveys?	This has been added to the supplementary material file.

Was the site being observed by hydroacoustics during the gas escape observed by the PZM?	No, unfortunately we did not acquire hydroacoustic data during piezometer deployment because the main focus was calypso coring and the whole schedule was against time.
General: Thinking more broadly, would it be possible/appropriate to apply the new information provided by the PZMs to the gas emission estimates for the Vestnesa Ridge or broader western Svalbard area to provide an example of how this information can be used? I expect the short answer to this is no, but this final section is missing an element of looking forward towards this type of goal to add perspective to the “significantly affect” and “significantly reduce” of the final two sentences. This comment links back to that above related to the final statement of the introduction section.	With only two piezometer deployments over 3 to 4 days the answer to the first question is no. We have added the following paragraph describing the way to integrate our analyzes into a more global approach to the last section. “Our results show that monitoring pore pressure in near-surface sediment is an approach that allows to constrain seafloor gas emissions and their governing processes beyond the limited capabilities of hydro-acoustic surveys. However, it is important to highlight the limitations of fixed-point observations with respect to the spatial coverage of a hydro-acoustic survey. Both methods are complementary and combining hydro-acoustic data and in-situ piezometer seems to be an effective approach to significantly reduce the uncertainties of emission rates due to temporal variability. We envision as next step the installation of long-term piezometer observatory offshore Svalbard to acquire long-time data series combined with recurrent hydro-acoustic surveys to further test our hypothesis and upscale our initial results for the broader Arctic. Such an approach is expected to improve predictive models of seabed gas emissions due to sea-level rise.”
Methods	
check consistency with use of present and future verb tense	Considered

REVIEWERS' COMMENTS

Reviewer #1 (Remarks to the Author):

The authors considered all of my recommendations. The manuscript is now in good shape and can be published in its present form without further revisions.

Reviewer #2 (Remarks to the Author):

The revised manuscript is substantially improved and provides a much more clear and therefore compelling discussion of the very interesting piezometer dataset and its implications.

Minor comments:

Abstract L30: "sea-level rise, however, seems to influence, [...]" The (long-term) process of sea level rise has not been observed, just the short term changes in sea pressure related to tide. This is clear in the preceding sentences and the next sentence, but suggest changing this to "Pressure changes related to water depth" or similar, thus explicitly stepping from what was observed (tides) to the wider implication (sea level) via the process (pressure).

Summary line 354: "strikingly correlated": suggest something more like "strikingly coincident" – no correlation is calculated and the text as written does not communicate that the 'correlation' is referring to the two things happening at the same time.

Supplemental Figure 1: "The shaded area marks the gas hydrate stability zone based on the presence of gas- hydrate related bottom simulating reflections in seismic profiles". I don't see a shaded area on the map (?) is this missing/could it be made more clear?

Supplemental Figure 6: add scale bar for improved context/comparison to other maps in this document. Were there no active gas flares observed during the (limited times) for which sonar data were collected in the areas shown? The absence of indication of their observation implies this is the case but it is not explicitly stated. I understand that full analysis of the sonar datasets is likely beyond the scope of this study, but as shown this figure is difficult to interpret/understand.

Associate Editor	
Comments	Reply
Main text	The manuscript complies now with the journal's format requirements. See annotated file.
Supplementray material	Corrected as suggested. See annotated file.
Reviewer #1	
Comments	Reply
The authors considered all of my recommendations. The manuscript is now in good shape and can be published in its present form without further revisions	The authors thank the referee for this positive feedback
Reviewer #2	
Comments	Reply
The revised manuscript is substantially improved and provides a much more clear and therefore compelling discussion of the very interesting piezometer dataset and it's implications.	We are happy about this and thank the reviewer for this positive assessment
Abstract L30: "sea-level rise, however, seems to influence, [...]" The (long-term) process of sea level rise has not been observed, just the short term changes in sea pressure related to tide. This is clear in the preceding sentences and the next sentence, but suggest changing this to "Pressure changes related to water depth" or similar, thus explicitly stepping from what was observed (tides) to the wider implication (sea level) via the process (pressure).	Considered and rephrased "Sea-level rise, however, seems to influence" replaced by "High tides, however, seem to influence"
Summary line 354: "strikingly correlated": suggest something more like "strikingly coincident" – no correlation is calculated and the text as written does not communicate that the 'correlation' is referring to the two things happening at the same time.	Considered and rephrased "fluctuations are strikingly correlated to tidal cycles" Replaced by "fluctuations are strikingly coincident with tidal cycles"
Supplemental Figure 1: "The shaded area marks the gas hydrate stability zone based on the presence of gas-hydrate related bottom simulating reflections in seismic profiles". I don't see a shaded area on the map (?) is this missing/could it be made more clear?	Corrected by reducing the transparency of the shaded area.
Supplemental Figure 6: add scale bar for improved context/comparison to other maps in this document. Were there no active gas flares observed during the (limited times) for which sonar data were collected in the areas shown? The absence of indication of their observation implies this is the case but it is not explicitly stated. I understand that full analysis of the sonar datasets is likely beyond the scope of this study, but as shown this figure is difficult to interpret/understand.	Considered by adding scale bar to figure 6. We added to the legend that "No gas seep was detected during the EM302 and EK60 sonar data acquisition"